# Visual Event-Related Potentials under External Emotional Stimuli in Bipolar I Disorder with and without Hypersexuality

**DOI:** 10.3390/brainsci12040441

**Published:** 2022-03-25

**Authors:** Chu Wang, Lars M. Rimol, Wei Wang

**Affiliations:** 1The Second Clinical Medical College, Zhejiang Chinese Medical University, Hangzhou 310053, China; 21718608@zju.edu.cn; 2Department of Clinical Psychology and Psychiatry, School of Public Health, Zhejiang University College of Medicine, Hangzhou 310058, China; 3Department of Psychology, Norwegian University of Science and Technology, 7491 Trondheim, Norway; lars.rimol@ntnu.no

**Keywords:** bipolar I disorder, electroencephalography (EEG), emotional stimuli, erotica, event-related potentials, hypersexuality, mood disorders, personality

## Abstract

Hypersexuality is related to functions of personality and emotion and is a salient symptom of bipolar I disorder especially during manic episode. However, it is uncertain whether bipolar I disorder with (BW) and without (BO) hypersexuality exhibits different cerebral activations under external emotion stimuli. In 54 healthy volunteers, 27 BW and 26 BO patients, we administered the visual oddball event-related potentials (ERPs) under external emotions of Disgust, Erotica, Fear, Happiness, Neutral, and Sadness. Participants’ concurrent states of mania, hypomania, and depression were also evaluated. The N1 latencies under Erotica and Happiness were prolonged, and the P3b amplitudes under Fear and Sadness were decreased in BW; the P3b amplitudes under Fear were increased in BO. The parietal, frontal, and occipital activations were found in BW, and the frontal and temporal activations in BO under different external emotional stimuli, respectively. Some ERP components were correlated with the concurrent affective states in three groups of participants. The primary perception under Erotica and Happiness, and voluntary attention under Fear and Sadness, were impaired in BW, while the voluntary attention under Fear was impaired in BO. Our study indicates different patterns of visual attentional deficits under different external emotions in BW and BO.

## 1. Introduction

Mood disorder is a psychiatric illness being predominated in emotional and cognitive disturbances [1,2], and its main type is bipolar disorder, a lifelong and recurrent disease being characterized by fluctuant mood and energy episodes of mania/hypomania and depression [3]. Bipolar disorder affects over 1% of the globe population, with high incidence rates of disability and suicide [4]. Its exact pathogenesis is not fully known, but nearly 70% of it is heritable [5]. Other contributing factors to its onset might be the monoaminergic dysregulation, inflammatory disturbance, and adverse environmental exposure [5,6,7]. In clinics, the precision diagnosis and management of bipolar disorder and its subtypes remain difficult after tremendous efforts these years, which might be due to the nonspecific symptoms or a depressive episode at the beginning and to the substantial psychiatric and somatic comorbidities [3,4].

For instance, the bipolar I disorder (BD I) has prominent impulsivity, irritability, and distractibility [8], and is characterized by the recurrent mania and depression episodes [9] which are subjected to severe dysfunctions of emotional processing and regulation [10,11]. Patients with BD I experienced great negative affectivity in their daily life [12], and showed impairment in recognizing facial happiness and disgust [13]. Interestingly, deficits in cognition, attention, and executive function are prominent in BD I which share a close relationship with the abnormal cerebral activities [14,15]. These activations might be motivated by two neural systems of attention: one is the top-down network involving the dorsal posterior parietal and frontal cortices, responsible for the goal-directed selection of stimuli, the other is the bottom-up network which is mediated by temporoparietal and ventral frontal cortices, in charge of detection for salient or unexpected stimuli [16]. The inhibitory control deficit which might be influenced by emotional stimuli is also a core feature of BD I [17,18], and this deficit in BD I is mainly correlated with the functional abnormalities involved in the right inferior frontal gyrus and the right superior temporal gyrus [19]. These neurobiological findings partly support and extend the crucial role of prefrontal areas (especially the inferior frontal gyrus and pre-supplementary motor area) played in the neural network of inhibitory control [20,21]. In addition to its attention and inhibitory control, the prefrontal cortex performs a vital role in the emotional generation and regulation, and in the neural circuitry underlying human learning regarding fear conditioning which contributes to the understanding of psychiatric disorders [22,23].

Hypersexuality which is characterized by intense and excessive sexual cognitions, fantasies, urges, or behaviors, is one of the best-known and influential symptoms of BD I [24]. It is also connected closely with emotional instability and impulsivity, which are often found in personality disorders [25,26]. Patients with Cluster B personality disorders, for example, have some hypersexuality features [27,28], and the difficulties of emotional regulation are suggested to be an etiological factor of hypersexuality in some psychiatric disorders such as BD I [29,30]. However, there is no direct evidence showing the differently emotional processes between BD I with (BW) and without (BO) hypersexuality. Considering the core features of hypersexuality [31,32], we might speculate that BW engages in more sexual-related or risky behaviors and experiences more negative emotions than BO.

Previous studies have shown that the attentional or cognitive functions of BD I are modulated by external emotions [33,34]. Attentional bias was when an individual was facing diversely external stimuli where the emotional stimuli received more attentional resources than the neutral ones [35]. Processing emotional stimuli mainly consists of the primary perception and the voluntary processing with attention and cognition deployment [36]. Moreover, attentional and cognitive deficits of external emotions in BD I patients vary due to their concurrent mood states. Previous studies have shown that the difficulties in recognizing facial fear and disgust were pronounced in BD I patients during manic episodes [37], and those in depressive episodes poorly recognized positive facial emotions and overidentified negative ones [38]. In manic phases, BD I patients displayed happiness-related bias during initial orienting and threat-related bias during attentional engagement in an eye-tracking study [39]. In an event-related potential (ERP) study, the P3 amplitudes triggered by negative facial emotions were decreased in manic patients compared to healthy volunteers [40], and BD I showed hyperactivation in the left insula coupled with hypoactivation in right supramarginal gyrus and left precuneus when exposed to emotionally negative images [11]. The increased activations of anterior cingulate, dorsolateral prefrontal cortex, and amygdala to facial fear and happiness also reported in BD I [10]. Moreover, BW and BO patients differently displayed the peripheral physiological responses to external emotions and their transitions, for instance, erotica and sadness [41]. However, clear pictures of cerebral attentional function in BW and BO under external emotional stimuli are still unclear.

With excellent temporal resolution, the ERPs serve as a precise index of cortical activity and are commonly used in detecting attentional and cognitive variations of human beings [42]. The ERP N1 and P2 components are involved with primary perception of incoming information and early allocation of attention to stimuli [43,44], respectively. N2 reflects the involuntary identification and distinction to deviant stimuli [45]. P3 component is involved in central resource allocation and voluntary processing of attention and working memory; its subcomponent P3a is related to stimulus-driven attention processing, and its P3b is associated with context updating and subsequent memory storage [46]. In the current study, we would like to use visual ERPs under externally emotional stimuli in BW and BO patients. We have hypothesized that: (1) compared to BO, BW would display more significant abnormalities in ERP components representing primary perception or involuntary attention under positive emotions, and in those representing voluntary attentional processing under negative emotions; (2) both BW and BO would have different patterns of cerebral activation under different external emotions; and (3) the ERP morphologies would correlate with the concurrently affective states of BW and BO.

## 2. Materials and Methods

### 2.1. Participants

Altogether 107 participants were recruited from the university, community, or psychiatric clinic: 54 healthy volunteers (21 males and 33 females; mean age: 19.56 years ± 2.00 S.D., age range: 18–26 years); 27 BW (17 males and 10 females; mean age: 19.22 ± 1.70, age range: 18–24); and 26 BO (10 males and 16 females; mean age: 19.85 ± 1.97, age range: 18–24) patients. Patients were diagnosed by an experienced psychiatrist (W.W.) according to DSM-5 diagnostic criteria for BD I [1] and the proposed diagnostic criteria for hypersexuality [29], together with the following questionnaires measuring affective states. There were no significant differences among the three groups as to gender (χ^2^ = 4.79, df = 2, *p* = 0.091), age (F (2, 104) = 0.70, mean square effect (MSE) = 2.59, *p* = 0.498), or education level (χ^2^ = 1.33, df = 2, *p* = 0.513). Through a semi-structured clinical interview, all participants were confirmed to be medication-free for at least one week and have no other confounding factors including schizophrenia, personality disorder, substance abuse, head injury, or central nervous system, and neurocognitive disorders. When necessary, two staff members were available to assist participants in filling in the demographic information, questionnaires, and completing the ERP tests. The study protocol was approved by a local ethics committee, and all participants had provided their written informed consent before participating in this study.

### 2.2. Questionnaires

All participants were asked to complete the three self-assessment questionnaires below in a quiet room, using a paper-and-pencil style.

A. The Mood Disorder Questionnaire (MDQ; [47]) is an instrument with 13 dichotomous items (yes/no) assessing mania/hypomania symptoms and behaviors, and two items evaluating the frequency of those symptoms and the extent of functional impairment. The internal reliability of the 13 dichotomous items was 0.77 in the current study.

B. The Hypomania Checklist-32 (HCL-32; [48]) comprises 32 items for detecting hypomanic symptoms. There are 32 dichotomous items (yes/no) regarding emotions, thoughts, or behaviors of hypomania, and other items about the duration, impact on family, social as well as work life, or other people’s reactions. Its internal reliability was 0.76 in the current study.

C. The Plutchik–van Praag Depression Inventory (PVP; [49]) consists of 34 items describing depression symptoms. Three scale points (0, 1, 2) of each item are corresponding to increasing tendencies of depression. If participants score between 20 and 25, they are considered to have “possible depression”, or “depression” if they score above 25. The internal reliability of this inventory was 0.82 in the current study.

### 2.3. ERP Designs and Recordings

#### 2.3.1. External Emotional Stimuli

The external emotional stimuli were composed of pictures selecting from the International Affective Picture System [50] and sounds from the International Affective Digital Sounds [51] of the same domain. The six scenes of distinct emotions, namely Disgust (picture code: 9325; sound code: 255), Erotica (4680; 205), Fear (3053; 275), Happiness (2040; 110), Neutral (5390; 172), and Sadness (2205; 295), were presented by eevoke^TM^ software 3.1 (ANT Software B.V., Enschede, The Netherlands). In each emotional scene, a color picture was horizontally presented (768 × 512 pixels), sustaining about 19.8° × 13.5° of visual angles. Simultaneously, a sound of 40–50 dB in intensity was delivered through headphones.

#### 2.3.2. ERP Paradigm

After completing the above-mentioned questionnaires, participants were led to a dimly lit room and seated at 100 cm from a computer screen. Six successive sessions (Disgust, Erotica, Fear, Happiness, Neutral, or Sadness) with a two-minute interval between adjacent sessions were randomly presented for each participant. Within each session, a fixation cross in the middle of a black background were presented for 3000 ms, followed by 150 ERP trials (each trial lasting for 2400 ms), with an inter-trial interval of 1200~1500 ms. Within each trial, an external emotional stimulus of either Disgust, Erotica, Fear, Happiness, Neutral, or Sadness was shown for 2000 ms; then either a standard (a square of 40 mm × 40 mm, lasting for 400 ms) or target (a circle of 40 mm in diameter, 400 ms) stimulus appeared in the middle of the black background (Figure 1). In a randomized order, the standard stimuli were delivered 120 times (80%) and the target stimuli 30 times (20%). Participants were instructed to respond to the target stimuli actively by pressing a button with their right index finger as soon as possible, and no reaction was needed to the standard stimuli or emotional displays.

#### 2.3.3. ERP Recording

EEG signals were recorded with 32 channel elastic electrocap (Electro-Cap International, Inc., Eaton, OH, USA) according to the 10–20 International System. The impedance of each electrode was maintained below 10 kΩ, and EEG signals were amplified by a DC amplifier (the ANT amplifier, Enschede, The Netherlands) with a sampling rate of 1024 Hz. Bipolar recordings of the electro-ocular activity were collected by electrodes placed at the outer canthus and supraorbitally to the right eye. Referred to the average activity of the two mastoid electrodes (M1 and M2), potentials were analyzed offline using a band-pass of 0.01~30 Hz in ASA software 4.7.3 (ANT Software B.V., Enschede, The Netherlands). The sampling epoch was 100 ms pre-stimulus and 600 ms post-stimulus. Any sweep in which the EEG exceeded ±70 μv or with electro-ocular activity was excluded from averaging.

Nine electrodes in frontal, central and parietal sites, i.e., F3, Fz, F4, C3, Cz, C4, P3, Pz, and P4 were selected, and ERP morphology determined by target stimuli were analyzed in terms of peak latency and baseline-to-peak amplitude. Latency ranges of potentials were 70~200 ms for N1, 150~300 ms for P2, 210~390 ms for N2, 300~500 ms for P3a, and 400~580 ms for P3b. Moreover, the reaction times and hit accuracies in response to target stimuli were recorded.

### 2.4. Statistical Analyses

One-way ANOVA was applied to the scale scores of MDQ, HCL-32, and PVP as well as reaction times in the three groups of participants. The latencies and amplitudes of ERP component in the three groups were analyzed by two-way ANOVA, i.e., group (3) × electrode (9). Whenever a significant main effect was detected, the Bonferroni test was employed as a post-hoc comparison. The *p* < 0.05 at no less than three coaxial electrodes (frontal, central, posterior in lateral axis; left, midline, right in sagittal axis) were considered to be significant and meaningful for group companions. Relationships between ERP components and questionnaire scores were examined using the Pearson correlation test, and only significant correlations with *p* < 0.01 at no less than three coaxial electrodes were considered as stable and meaningful.

The respective 3D sources were reconstructed based on data obtained at the 32 electrodes, to observe the involvement of cerebral areas corresponding to the significant differences of target stimuli under specific external emotional stimuli in the three groups. The source reconstruction applied the SPM12 software package, running in Matlab R2014b (Mathworks Inc., Natick, MA, USA). The figures of the averaged source map were generated by xjView 10.0 (http://www.alivelearn.net/xjview (accessed on 18 February 2022)).

## 3. Results

### 3.1. Concurrent Affective States and Behavioral Results

The MDQ scores were significantly different among the three groups of participants (F (2, 104) = 30.89, *p* < 0.001, MSE = 198.04), with BW (*p* < 0.001, 95% confidence interval (CI) = 2.68~5.58) and BO (*p* < 0.001, 95% CI = 2.03~4.97) scored higher than controls did. The HCL-32 scores were also significantly different among the three groups (F (2, 104) = 64.89, *p* < 0.001, MSE = 521.55). The BW (*p* < 0.001, 95% CI = 4.95~8.20) and BO (*p* < 0.001, 95% CI = 4.22~7.51) reported higher hypomania symptoms than controls did (Table 1). No significant difference was found in PVP scores (F (2, 104) = 0.73, *p* = 0.486, MSE = 32.31), reaction times (F (2, 95) = 0.50~2.13, *p* = 0.125~0.609, MSE = 3347.00~17,572.70), or reaction accuracies (F (2, 95) = 0.19~2.35, *p* = 0.101~0.831, MSE = 9.31~109.93).

### 3.2. ERP Components

The ERP components N1, P2, N2, P3a, and P3b under differently external emotions were collected in three groups of participants. For the sake of brevity, only data showing significant differences between groups were reported here, all other data are available upon request. There were significant differences on N1 latencies under Erotica (group effect, F (2, 104) = 7.18, *p* = 0.001, MSE = 31,764.79; electrode effect, F (8, 832) = 13.59, *p* < 0.001, MSE = 2638.88; group × electrode effect, F (16, 832) = 1.04, *p* = 0.410, MSE = 202.12) and Happiness (group effect, F (2, 103) = 3.96, *p* = 0.022, MSE = 15,955.06; electrode effect, F (8, 824) = 6.51, *p* < 0.001, MSE = 1514.16; group × electrode effect, F (16, 824) = 0.92, *p* = 0.548, MSE = 213.52) among the three groups of participants. N1 latencies under Erotica at frontal and central electrodes were prolonged in BW compared to those in BO (*p* = 0.002~0.042) and controls (*p* = 0.000~0.036), at left electrodes were prolonged in BW (*p* = 0.002~0.045) compared to those in BO, and at middle electrodes were prolonged in BW (*p* = 0.000~0.008) compared to those in controls. N1 latencies under Happiness at frontal electrodes were prolonged in BW (*p* = 0.000~0.045) compared to those in controls. The P3b amplitudes under Fear (group effect, F (2, 103) = 7.94, *p* = 0.001, MSE = 929.93; electrode effect, F (8, 824) = 11.10, *p* < 0.001, MSE = 29.75; group × electrode effect, F (16, 824) = 1.61, *p* = 0.060, MSE = 4.32) and Sadness (group effect, F (2, 103) = 3.73, *p* = 0.027, MSE = 478.13; electrode effect, F (8, 824) = 9.22, *p* < 0.001, MSE = 32.51; group × electrode effect, F (16, 824) = 1.33, *p* = 0.171, MSE = 4.69) in three groups were also statistically different: P3b amplitudes under Fear at all nine electrodes (*p* = 0.001~0.011) and under Sadness at middle electrodes in BW (*p* = 0.005~0.048) were decreased than those in BO, and P3b amplitudes under Fear at all nine electrodes in BO (*p* = 0.001~0.025) were increased than those in controls (Table 2). The P2 latencies under Sadness (group effect, F (2, 103) = 4.04, *p* = 0.020, MSE = 24,782.99; electrode effect, F (8, 824) = 4.37, *p* < 0.001, MSE = 1627.81; group × electrode effect, F (16, 824) = 1.34, *p* = 0.165, MSE = 499.23) and P3b amplitudes under Erotica (group effect, F (2, 104) = 3.13, *p* = 0.048, MSE = 265.95; electrode effect, F (8, 832) = 5.26, *p* < 0.001, MSE = 14.26; group × electrode effect, F (16, 832) = 0.927, *p* = 0.538, MSE = 2.51) were also statistically different among three groups of participants. However, the post-hoc comparisons did not detect any meaningfully between-group differences. As an example, the grand averages of ERPs under external Erotica at nine electrodes in three groups were presented in Figure 2. The differences of N1 latencies under Erotica and Happiness, and P3b amplitudes under Fear and Sadness at Cz in three groups were illustrated in Figure 3.

### 3.3. Source Reconstructions

After significant differences on N1 latencies and P3b amplitudes under different external emotions were found, we located the possible neural sources for these components by performing 3D source reconstruction in 70–200 ms and 400–580 ms time-windows in three groups, respectively. In controls, the bilateral supramarginal gyri under Erotica and right inferior temporal gyrus under Happiness were mainly activated in N1 time windows. During P3b time window, the right inferior occipital gyrus under Fear and bilateral medial frontal gyri were activated. In BW, enhanced processing of left postcentral gyrus, right supramarginal gyrus, and bilateral inferior frontal gyri were found during Erotica and Happiness in N1 time window, and of bilateral lingual gyri as well as left medial frontal gyrus during Fear and Sadness in P3b time window. In BO, the bilateral medial frontal gyri and right inferior temporal gyrus were processed under Erotica in N1 time window and Fear in P3b time window, while the left superior/inferior frontal gyri were involved with Happiness during N1 time window and Sadness during P3b time window (Table 3). As an example, the 3D source reconstruction of N1 to Erotica in three groups was shown in Figure 4

### 3.4. Relationships between ERPs and Concurrent Affective States

In controls, the N1 amplitudes under Sadness in frontal electrodes were positively correlated with MDQ (*n* = 54, r = 0.38~0.39, *p* = 0.003~0.005). In BW, the P3a amplitudes under Erotica in all nine electrodes except for C3 were positively correlated with PVP (*n* = 27, r = 0.50~0.60, *p* = 0.001~0.008). In BO, the N1 amplitudes under Fear in right electrodes were negatively correlated with HCL-32 (*n* = 25, r = −0.53~−0.58, *p* = 0.002~0.007). No other meaningful relationship between ERP components and affective states was found in any given group. Taking an example, the correlation between P3a amplitude (Cz) under Erotica and the PVP score in BW was displayed in Figure 5.

## 4. Discussion

To the best of our knowledge, this is the first study examining visual ERPs under external emotions in BW and BO. Confirming our hypotheses, we found higher MDQ and HCL-32 scores in both BW and BO than those in controls, which is in accordance with former studies showing that BD I exhibit high levels of manic or hypomanic symptoms [41,52]. N1 latencies under Erotica and Happiness were prolonged in BW when compared to BO or healthy controls. P3b amplitudes were decreased under Fear and Sadness in BW compared to those in BO, and were increased in BO under Fear compared to controls. Abnormal activation of the cerebral areas especially frontal regions was detected corresponding to N1 and P3b components under different external emotions. Moreover, the concurrent affective states were correlated with ERPs in three groups.

In healthy controls, processing in Erotica and Happiness during N1 time window were mainly activated the bilateral supramarginal gyri and right inferior temporal gyrus, respectively. This might be supported by the connection of temporo-parietal lobe under positive emotion [53]. The right inferior occipital gyrus and bilateral medial frontal gyri were activated in Fear and Sadness during P3b time window, which was consistent with the previous studies showing that the neural activations of negative emotion were in right occipital gyrus, right amygdala, and bilateral frontal regions [54,55]. The correlation between mania and the elementary encoding of Sadness might be explained by the fact that the manic patients reported attenuated subjective sensation of facial sadness [56].

In BW, N1 latencies under Erotica and Happiness were prolonged. Since N1 reflects the primary processing of incoming information [44], the prolonged N1 latencies under Erotica and Happiness implied a delayed processing of positive emotions. Hypersexual individuals exhibited fewer positive emotions and displayed a paucity of happiness [57], which might influence the recognition of happiness stimuli. The differences of N1 latencies under Erotica between BW and BO might be due to the compulsive use of sexual materials in BW [31], thus leading to a decrease of emotional sensitivity to erotica. On the other hand, the hypersexual behaviors were associated with negative emotions [57], and the impaired recognition of unpleasant emotions in BD I [13,37] might result in the delayed perception of Erotica in BW in the current study. Through source analyses, we also found different parts of the brain activated under external erotica and happiness in BW. The literature shows that the postcentral gyrus plays a critical role in processing sensory information and regulation of emotion [58], and the supramarginal gyrus is a part of the ventral attention network [59] associated with overcoming emotional egocentric biases [60]. The inferior frontal gyrus is activated during attentional control and emotional perception [61,62]. In addition to encoding the primary perception of stimuli, the activations of these regions might imply that Erotica rather than Happiness induces the activation of somatosensory areas in BW. The finding that negative and distressing emotions triggered excessive sexual behaviors [63] helps explain the positive correlation between PVP and P3a amplitudes under Erotica in our BW patients.

In BO, P3b amplitudes under Fear were increased, suggesting the heightened voluntary attention to the negative emotions in these patients, which might be due to difficulties of ignoring the threating stimuli during attentional engagement [64]. The decreased P3b amplitudes in BW rather than in BO under Fear and Sadness might be explained by the following documentation. Hypersexuality is a typical symptom of mania episode [24], patients with mania displayed an impaired recognition of facial fear and sadness [56,65], and displayed the emotional dysfunction manifested in hypersexuality [26]. These abnormalities might lead to insufficient voluntary attention and evaluation to Fear and Sadness in BW. On the other hand, the neural regions activated during Fear were diverse in BW and BO during P3b time window. Indeed, bilateral lingual gyri associated with visual memory and stimuli perception [66] were activated in BW. Interestingly, bilateral medial frontal gyri which play a critical role in executive function, decision making, and cognitive control of emotion [67,68], and right inferior temporal gyrus, which correlates with object recognition [69], were mainly activated in BO. Moreover, the N1 amplitudes under Fear in right electrodes were negatively correlated with HCL-32 in BO. In clinics, individuals with hypomania show elevated mood, increased energy or activity, flights of ideas, and racing thought [9], which might facilitate the primary recognition of fearful or threating scenes in BO.

Similar to our present findings, Sagar et al. [10] showed that BD I had similar activated brain regions under fearful and happy stimuli. More specifically, we discovered that even during different time windows, activations of prefrontal cortex were altered in BW and BO under both Happiness and Sadness. Indeed, the activation of prefrontal cortex is critical in the emotional regulation [23]. In our BO, there were more activations of bilateral medial frontal gyri and right inferior temporal gyrus under both Erotica and Fear, which might imply that the primary perception under Erotica and the voluntary attention under Fear were more contrasted to distinguish BW and BO than those under Happiness and Sadness.

Considering the disruption of cognitive and cerebral processing under external emotions in BW and BO, the non-invasive brain stimulation might be considered as an effective approach to enhance neurocognition and regulate behaviors of the two BD I subtypes through applying to the specific brain areas, such as the prefrontal cortex which ties closely with cognition and emotional regulation [70]. Indeed, this stimulation method has been adopted in the management of psychiatric and neurological diseases [70,71], in improvement of emotion recognition [72], and in modulation of memory, particularly fear-related ones [73,74]. For the current challenges of precision diagnosis and management, this stimulation method within brain functional regions is potentially applicable in the individualized treatment of psychiatric problems including bipolar disorder.

## 5. Limitations and Future Directions

Nonetheless our study suffers from several design limitations. Firstly, our participants were all young adults, whether the results can be generalized to other age groups remains to be seen. Secondly, we only enrolled BW and BO patients, recruiting other groups such as major depression or personality disorder might add more confirmation to our current findings. Thirdly, we did not include other external emotions such as surprise, anger, or contempt, which might also display their emotional effects on the attentional process. Fourthly, we failed to measure personality traits in our participants, since BD I and hypersexuality might be related to either normal or disordered personality traits. Fifthly, we did not follow up to the eventual therapies of our patients, which might affect their sexuality in a far-reaching way. Nevertheless, we have demonstrated that BW had delayed primary perception under Erotica and Happiness and decreased voluntary attention under Fear and Sadness, and demonstrated differences between BW and BO on attentional function under Erotica and Fear. Thus, our study might provide hints of different emotional processes of and the clinical intervention for the two subtypes of bipolar I disorder.

## 6. Conclusions

We have found that BW and BO demonstrated different cerebral processing and activations under external emotions, especially Erotica and Fear. Our study thus indicates different patterns of visual attentional deficits and emotional processes and provides a basis for developing emotional intervention therapy and applying the non-invasive brain stimulation to brain areas including prefrontal cortex in the two bipolar I disorder subtypes.

## Figures and Tables

**Figure 1 brainsci-12-00441-f001:**
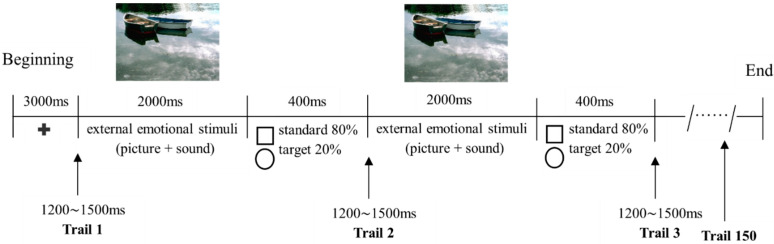
Timeline of events in the oddball paradigm. Participants were instructed to respond to a circle picture (target stimulus) as quickly as possible by pressing a button with their right index finger, and to do nothing to the square picture (standard stimulus) or emotional scenes (taking Neutral as an example). Note: **+**, fixation cross representing the beginning of the paradigm.

**Figure 2 brainsci-12-00441-f002:**
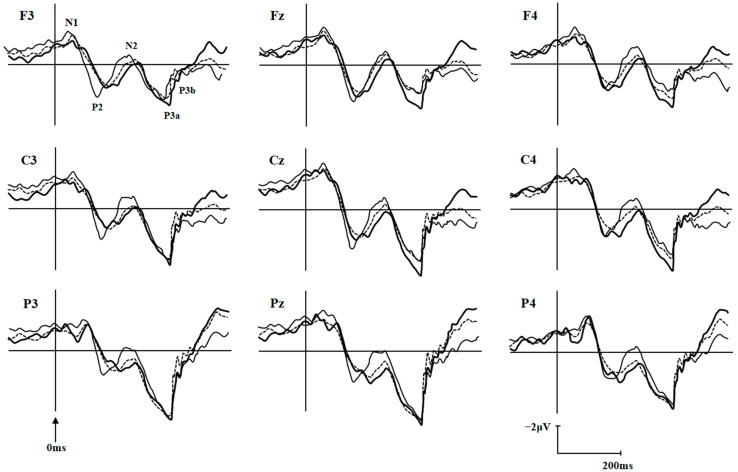
Grand averages of ERPs elicited by target under external emotional condition of Erotica at nine electrodes in the healthy volunteers (dashed line; *n* = 54), bipolar I disorder patients with (thick line; *n* = 27) and without (thin line; *n* = 26) hypersexuality.

**Figure 3 brainsci-12-00441-f003:**
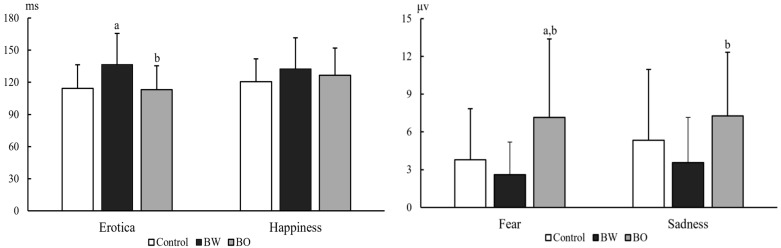
N1 latencies under Erotica and Happiness (left graph), and P3b amplitudes under Fear and Sadness (right graph) at Cz in healthy volunteers (Controls, *n* = 54), bipolar I disorder patients with (BW, *n* = 27) and without (BO, *n* = 26) hypersexuality. Notes: a, *p* < 0.05 versus controls; b, *p* < 0.05 versus BW.

**Figure 4 brainsci-12-00441-f004:**
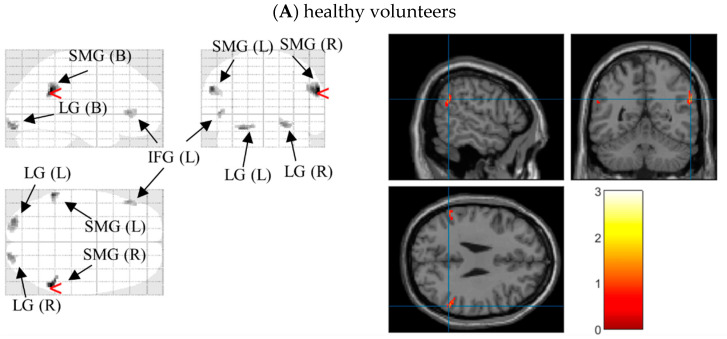
Source map of N1 (time window: 70~200 ms) to the target stimuli under Erotica in healthy volunteers (**A**), bipolar I disorder patients with (**B**) and without (**C**) hypersexuality (generated by xjView 10.0 (http://www.alivelearn.net/xjview (accessed on 18 February 2022)). Left panels show sources overlaid on a glass brain; Right panels show sources overlaid on a T1 template brain. The red arrowheads and blue cross hairs indicate source regions of peak intensity in the three groups were the right supramarginal gyrus, left postcentral gyrus, and right medial frontal gyrus, respectively. Notes: SMG (B), bilateral supramarginal gyri; LG (B), bilateral lingual gyri; IFG (L), left inferior frontal gyrus; PCG (L), left postcentral gyrus, SMG (R), right supramarginal gyri; IFG (R), right inferior frontal gyrus; MFG (B): bilateral medial frontal gyri; ITG (R), right inferior temporal gyrus. The bar in each panel on the right side indicates the intensity of cerebral activities, with white as the highest and black the lowest.

**Figure 5 brainsci-12-00441-f005:**
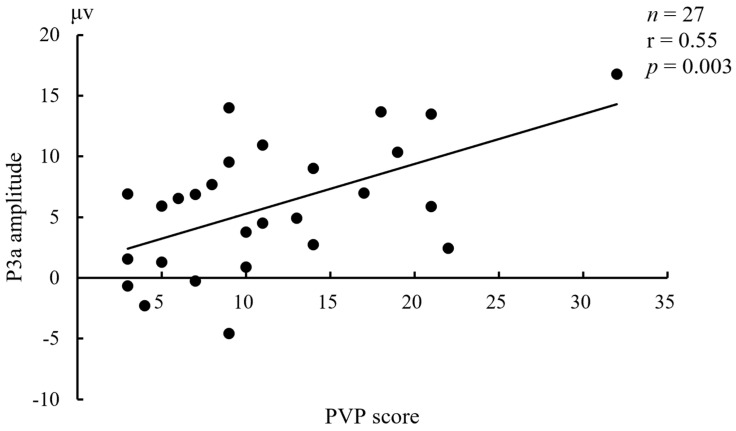
Relationship between P3a amplitude (Cz) under Erotica and the Plutchik–van Praag Depression Inventory (PVP) score in bipolar I disorder patients with hypersexuality.

**Table 1 brainsci-12-00441-t001:** Mental health symptoms, reaction times, and response accuracies to target stimuli (mean ± SD) in the healthy volunteers (Controls, *n* = 54), bipolar I disorder patients with (BW, *n* = 27) and without (BO, *n* = 26) hypersexuality.

	Controls	BW	BO
MDQ	5.61 ± 3.19	9.74 ± 1.70 a	9.12 ± 1.45 a
HCL-32	15.91 ± 3.42	22.48 ± 2.47 a	21.77 ± 1.53 a
PVP	13.30 ± 6.62	11.52 ± 7.12	12.08 ± 6.26
Reaction time (ms)			
under Disgust	489.45 ± 85.85	501.94 ± 57.37	517.96 ± 104.08
under Erotica	516.68 ± 99.59	541.12 ± 69.88	536.45 ± 117.45
under Fear	496.19 ± 83.93	536.20 ± 88.27	531.39 ± 105.95
under Happiness	489.81 ± 82.42	516.56 ± 60.61	513.74 ± 100.08
under Neutral	489.06 ± 84.97	532.34 ± 83.78	512.94 ± 112.24
under Sadness	498.96 ± 88.48	513.14 ± 66.33	539.83 ± 111.03
Response accuracy (%)			
under Disgust	94.18 ± 7.00	97.04 ± 2.97	93.61 ± 7.67
under Erotica	97.23 ± 5.62	94.32± 12.67	97.22 ± 5.70
under Fear	97.59 ± 5.93	97.04 ± 6.36	96.53 ± 9.55
under Happiness	95.46 ± 13.05	96.42 ± 8.16	99.17 ± 2.25
under Neutral	98.30 ± 3.60	96.05 ± 7.90	98.75 ± 2.37
under Sadness	98.08 ± 4.95	95.06 ± 11.82	97.36 ± 3.68

Notes: a, *p* < 0.05 versus controls; MDQ, The Mood Disorder Questionnaire; HCL-32, The Hypomania Checklist-32; PVP, The Plutchik–van Praag Depression Inventory.

**Table 2 brainsci-12-00441-t002:** N1 latencies (mean ± S.D.) under Erotica and Happiness and P3b amplitudes under Fear and Sadness in the healthy volunteers (Controls, *n* = 54), bipolar I disorder patients with (BW, *n* = 27) and without (BO, *n* = 26) hypersexuality.

		Controls	BW	BO	95% CI		
					Control-BW	Control-BO	BW-BO
N1 latency (ms)							
Erotica	F3	115.59 ± 21.74	136.36 ± 30.50 a	115.50 ± 21.89 b	−34.69~−6.86	−14.00~14.18	4.65~37.09
	Fz	112.60 ± 20.29	131.84 ± 30.09 a	113.54 ± 24.65 b	−33.08~−5.40	−14.95~13.08	2.17~34.43
	F4	117.02 ± 22.49	135.10 ± 30.48 a	118.09 ± 22.52 b	−32.27~−3.89	−15.44~13.30	0.47~33.55
	C3	116.00 ± 23.88	138.71 ± 30.39 a	113.81 ± 24.16 b	−37.46~−7.96	−12.74~17.14	7.71~42.11
	Cz	114.34 ± 22.00	136.76 ± 28.96 a	113.24 ± 22.21 b	−36.17~−8.67	−12.83~15.02	7.49~39.55
	C4	123.33 ± 25.81	139.69 ± 29.40 a	119.06 ± 27.50 b	−31.94~−0.79	−11.51~20.04	2.47~38.78
	P3	130.15 ± 30.46	142.80 ± 28.71	123.08 ± 26.04 b	−29.30~3.99	−9.79~23.92	0.32~39.12
	Pz	120.43 ± 27.84	139.87 ± 27.90 a	124.81 ± 23.13	−34.81~−4.07	−19.94~11.19	−2.86~32.98
	P4	130.85 ± 26.04	144.32 ± 25.28	130.14 ± 25.51	−28.22~1.28	−14.23~15.65	−3.02~31.37
Happiness	F3	122.07 ± 19.91	138.39 ± 26.40 a	130.26 ± 25.47	−29.54~−3.10	−21.76~5.37	−7.44~23.69
	Fz	120.88 ± 19.58	134.63 ± 29.30 a	127.06 ± 24.80	−27.29~−0.21	−20.08~7.71	−8.38~23.51
	F4	119.95 ± 21.91	142.48 ± 23.73 a	131.20 ± 26.77	−36.05~−8.99	−25.13~2.64	−4.66~27.20
	C3	120.50 ± 23.36	133.54 ± 28.15	130.97 ± 27.00	−27.68~1.59	−25.49~4.55	−14.66~19.81
	Cz	120.64 ± 21.22	132.53 ± 29.17	126.51 ± 25.52	−25.78~2.01	−20.13~8.39	−10.35~22.38
	C4	121.49 ± 23.75	137.56 ± 28.23 a	134.01 ± 23.30	−30.32~−1.81	−27.15~2.11	−13.25~20.33
	P3	129.27 ± 27.59	140.09 ± 34.08	132.84 ± 25.38	−27.40~5.75	−20.59~13.44	−12.27~26.76
	Pz	127.82 ± 24.53	135.67 ± 32.15	131.79 ± 25.11	−23.22~7.51	−19.73~11.80	−14.20~21.98
	P4	134.24 ± 30.23	145.19 ± 30.88	140.30 ± 23.59	−27.58~5.69	−23.13~11.01	−14.70~24.47
P3b amplitude (μv)							
Fear	F3	2.30 ± 2.66	1.90 ± 2.80	4.55 ± 4.53 a,b	−1.45~2.25	−4.15~−0.35	−4.83~−0.47
	Fz	2.69 ± 3.08	1.47 ± 4.14	5.55 ± 5.24 a,b	−1.04~3.49	−5.18~−0.54	−6.75~−1.42
	F4	2.68 ± 2.66	1.93 ± 2.55	5.06 ± 4.88 a,b	−1.14~2.64	−4.32~−0.45	−5.36~−0.91
	C3	3.10 ± 3.39	2.51 ± 3.49	6.17 ± 5.04 a,b	−1.62~2.81	−5.34~−0.80	−6.27~−1.05
	Cz	3.78 ± 4.07	2.60 ± 3.50	7.14 ± 6.25 a,b	−1.43~3.79	−6.05~−0.69	−7.62~−1.47
	C4	3.31 ± 3.22	2.50 ± 3.11	5.69 ±4.88 a,b	−1.28~2.91	−4.52~−0.22	−5.65~−0.72
	P3	2.07 ± 3.69	1.72 ± 3.06	5.65 ± 5.62 a,b	−1.20~2.69	−5.99~−1.17	−6.69~−1.17
	Pz	2.55 ± 4.04	1.97 ± 4.59	6.55 ± 6.42 a,b	−2.19~3.35	−6.84~−1.16	−7.84~−1.32
	P4	1.92 ± 3.26	1.43 ± 2.88	4.85 ± 4.73 a,b	−1.56~2.54	−5.04~−0.83	−5.84~−1.01
Sadness	F3	3.90 ± 4.10	3.04 ± 2.76	5.25± 3.95	−1.30~3.02	−3.56~0.87	−4.75~0.34
	Fz	4.21 ± 4.73	2.62 ± 2.29	5.50 ± 4.65 b	−0.84~4.01	−3.78~1.20	−5.73~−0.02
	F4	3.87 ± 4.15	2.87 ± 2.13	5.71 ± 4.25 b	−1.17~3.16	−4.06~0.38	−5.38~−0.29
	C3	4.60 ± 4.78	3.23 ± 2.63	6.33 ± 4.04 b	−1.02~3.75	−4.18~0.72	−5.91~−0.29
	Cz	5.34 ± 5.63	3.57 ± 2.51	7.28 ± 5.05 b	−1.03~4.57	−4.81~0.94	−7.01~−0.41
	C4	4.31 ± 4.82	3.60 ± 3.16	5.51 ± 5.43	−1.94~3.36	−3.92~1.52	−5.03~1.21
	P3	3.33 ± 4.13	2.43 ± 2.94	5.36 ± 4.49 b	−1.38~3.16	−4.37~0.30	−5.60~−0.25
	Pz	3.55 ± 4.36	2.72 ± 2.89	6.56 ± 5.17 a,b	−1.61~3.27	−5.51~−0.50	−6.71~−0.96
	P4	2.95 ± 4.02	2.52 ± 2.75	4.57 ± 3.92	−1.70~2.56	−3.81~0.57	−4.56~0.46

Notes: a, *p* < 0.05 versus controls; b, *p* < 0.05 versus BW; 95% CI, 95% confidence interval.

**Table 3 brainsci-12-00441-t003:** Putative N1 and P3b generators in the N1 and P3b time windows in healthy volunteers (Controls, *n* = 54), bipolar I disorder patients with (BW, *n* = 27) and without (BO, *n* = 26) hypersexuality.

Component	Group	Source Region
Erotica under N1 (70~200 ms)	Controls	supramarginal gyri (B) (Parietal Lobe) *
		lingual gyri (B) (Occipital Lobe)
		inferior frontal gyrus (L) (Frontal Lobe)
	BW	postcentral gyrus (L) (Parietal Lobe) *
		supramarginal gyrus (R) (Parietal Lobe)
		inferior frontal gyrus (R) (Frontal Lobe)
	BO	medial frontal gyri (B) (Frontal Lobe) *
		inferior temporal gyrus (R) (Temporal Lobe)
Happiness under N1 (70~200 ms)	Controls	inferior temporal gyrus (R) (Temporal Lobe) *
	BW	inferior frontal gyri (B) (Frontal Lobe) *
	BO	superior frontal gyrus (L) (Frontal Lobe) *
Fear under P3b (400~580 ms)	Controls	inferior occipital gyrus (R) (Occipital Lobe) *
	BW	lingual gyri (B) (Occipital Lobe) *
	BO	medial frontal gyri (B) (Frontal Lobe) *
		inferior temporal gyrus (R) (Temporal Lobe)
Sadness under P3b (400~580 ms)	Controls	medial frontal gyri (B) (Frontal Lobe) *
		inferior frontal gyrus (L) (Frontal Lobe)
		inferior temporal gyrus (R) (Temporal Lobe)
		superior frontal gyrus (R) (Frontal Lobe)
	BW	medial frontal gyrus (L) (Frontal Lobe) *
	BO	inferior frontal gyrus (L) (Frontal Lobe) *

Notes: *, source regions of peak intensity under different emotional stimuli in three groups; L, only the left side was activated; R, only the right side was activated; B, bilateral sides were activated.

## Data Availability

Data are available from the corresponding author (W.W.) upon reasonable request.

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
