# Peer review of "Visual Event-Related Potentials under External Emotional Stimuli in Bipolar I Disorder with and without Hypersexuality"

_brainsci, 2022, doi:10.3390/brainsci12040441_

Round 1

Reviewer 1 Report

Dear Authors,

Wang and colleagues in the present study entitled ‘Visual event-related potentials under external emotional stimuli in bipolar I disorder with and without hypersexuality’, investigated the current status of knowledge of differential brain activation for external stimuli in bipolar disorder with (BW) and without (BO) hypersexuality. For this purpose, in 54 healthy volunteers, 27 BW, and 26 BO patients, the visual oddball event-related potentials (ERPs) under external emotions of Disgust, Erotica, Fear, Happiness, Neutral and Sadness was administered. Participants’ concurrent states of mania, hypomania, and depression were also evaluated. Results showed that P3b amplitudes under Fear and Sadness conditions were decreased in BW, while the P3b amplitudes under Fear were increased in BO. Also, the parietal, frontal, and occipital activations were found in BW, and the frontal and temporal activations in BO under different external emotional stimuli respectively.

The main strength of this manuscript is that it addresses an interesting and timely question, providing a captivating interpretation and describing how different patterns of visual deficits under different external emotions are impaired in BW and in BO. In general, I think the idea of this perspective article is really interesting and the authors’ fascinating observations on this timely topic may be of interest to the readers of Brain Sciences. However, some comments, as well as some crucial evidence that should be included to support the author’s argumentation, needed to be addressed to improve the quality of the manuscript, its adequacy, and its readability, in particular reshaping parts of the Introduction and Discussion sections by adding more evidence.

Please consider the following comments:

  1. A graphical abstract summarizing the manuscript is highly recommended.
  2. Abstract: According to the Journal’s guidelines, the abstract should be a total of about 200 words maximum and should be introduced as a single paragraph, following the style of structured abstracts, but without headings. Please correct the actual one.
  3. Keywords: Please consider adding ‘Electroencephalography (EEG)’ and ‘Mood disorders’ as keywords and make it up to eight to ten keywords.
  4. In general, I recommend authors to use more evidence to back their claims, especially in the Introduction of the article, which I believe is currently lacking. Thus, I recommend the authors to attempt to deepen the subject of their manuscript, as the bibliography is too concise: nonetheless, in my opinion, less than 60/70 articles for a research paper are really insufficient. Indeed, currently authors cite only 55 papers, and they are too low. Therefore, I suggest the authors to focus their efforts on researching more relevant literature: I believe that adding more studies and reviews will help them to provide better and more accurate background to this study. In this review, I will try to help the authors by suggesting some relevant literature of my knowledge that suit their manuscript.
  5. Page 1, Introduction: I recommend opening the introduction with brief descriptions of mood disorder, bipolar disorders, and then bipolar I disorder, including pathogenesis, heterogeneity, comorbidity, precision diagnosis, treatment, and current challenge , among others (https://doi.org/10.3390/life11121365; https://doi.org/10.3390/biomedicines9070734; https://doi.org/3390/biomedicines9040403; doi: 10.1080/14728222.2020.1836160; https://doi.org/10.3390/medicina57080771), leading to the importance of subtypes, the rationale, and the purpose of this study. The authors take a narrow view of the cognitive impairment related to mood disorders, precisely focusing on difficulty in processing emotional stimuli in patients with depression/bipolar disorder. Nevertheless, I believe that a deeper examination of the mechanisms underlying impairments in emotion processing, and how this ability, together with deficient inhibitory control, are core factors in different psychopathologies, such as bipolar disorder. Interestingly, results from a recent review (https://doi.org/10.1016/j.brat.2021.103963) outlined typical dysfunctional behaviours, such as deficit in action control and motor inhibition, that are associated with psychopathological and psychiatric conditions, which are characterized by serious impulsivity problems that can determine significant impairment or distress (due to poor regulation and capacity of control, which can be intensified in the presence of emotional stimuli). Accordingly, another recent review, explored neural circuits of action inhibition in humans and the associated performance deficits in action control seen in various psychiatric disorders (https://doi.org/10.1016/j.cortex.2020.09.002).
  6. Materials and Methods: I suggest Authors to reorganize/rewrite this paragraph because, as it stands, this section is way too much inhomogeneous and dispersive, and describes the research procedures in an excessively broad way. To begin with, I suggest to clearly define diagnostic criteria and describe clinical tests utilized for the diagnosis of bipolar disorder, then provide complete information about ERP recording and analysis.
  7. Results: Please reorganize this section for clarity, providing full statistical information to ensure in-depth understanding and replicability of the findings. Also, please present statistical data in more detailed tables or figures.
  8. Discussion: In my opinion, this research article would be more compelling and useful to a broad readership if the authors moved beyond and discussed theoretical and methodological avenues in need of refinement, using this evidence to suggest a path forward.
  9. Discussion: Following the previous point raised, in order to provide a more though background on this topic, I would also suggest, a recent review that described the potential and effectiveness of non-invasive brain simulation (NIBS) to interfere and modulate the abnormal activity of neural circuits (i.e., amygdala-mPFC-hippocampus) involved in the acquisition and consolidation of fear memories, which are altered in many mood psychiatric disorders (i.e., bipolar disorder, anxiety disorder, specific phobias, post-traumatic stress disorder or depression) (https://doi.org/10.3390/ijms22052468; https://doi.org/10.1016/j.neubiorev.2021.04.036; https://doi.org/10.3390/biomedicines9070783 ).
  10. Even though it is not mandatory, I believe that the ‘Conclusions’ section would be useful to adequately indicate convey what the authors believe is the take-home message of their study, and therefore provide a synthesis of the data presented in the paper as well as possible keys to advancing research and understanding of the prevalence of depression in post-stroke patients.
  11. In according to the previous comment, I would ask the authors to better define a proper ‘Limitations and future directions’ section before the end of the manuscript, in which authors can describe in detail and report all the technical issues brought to the surface.
  12. Figures: I suggest modifying the Figure 3 for clarity because, as it stands, the readers may have difficulty comprehending it. In my opinion, authors should provide higher-resolution image of brain areas showed in the right panel, to allow a better detection of source regions of peak intensity. Also, please change the scale of the vertical axis and use the same minimum/maximum scale value in all the graphs.
  13. References: Authors should consider revising the bibliography, as there are several incorrect citations. Indeed, according to the Journal’s guidelines, they should provide the abbreviated journal name in italics, the year of publication in bold, the volume number in italics for all the references.

Overall, the manuscript contains 3 figures, 1 table and 55 references. In my opinion, the number of references it is low for an original research article, and this prevents the possibility of publishing it in this form – in my opinion. References should be more than 60/70 for original research articles. However, the manuscript might carry important value presenting how different patterns of visual deficits under different external emotions are impaired in BW and in BO.

I hope that, after these careful revisions, the manuscript can meet the Journal’s high standards.

I am available for a new round of revision of this article.

Best regards,

Author Response

Please see the attachment, thank you!

Reviewer 2 Report

Dear Editor,
I really appreciate the opportunity to review the manuscript brainsci-1625230 entitled:
"Visual event-related potentials under external emotional stimuli in bipolar I disorder with and without hypersexuality"

I commend the authors for describing this critical and timely issue. The paper is interesting and well-written; however, I would like to highlight some issues that merit revision, particularly I would like to ask the authors to specify in the part related to the methods what were the eventual therapies taken by the patients, which as it is known, affect in a not negligible way various aspects of sexuality.
If this data is not available, please, add a short paragraph in the limitations.

Author Response

Please see the attachment, thank you!

Reviewer 3 Report

Here are some suggestions to help improve the manuscript:

1) For the most significant results from Table 2, there should be plots made either as a bar plot or as a scatter plot showing the 3 groups and their values.

2) For some of the most significant correlations found in “section 3.4. Relationships between ERPs and concurrent affective states”: there should be scatter plot figures showing relationship between ERP and affective states complete with regression line.

3) Figure 3 legend should state the software technique that was used for source localization.

Author Response

Please see the attachment, thank you!

Round 2

Reviewer 1 Report

Manuscript ID: brainsci-1625230

Type: Article

Title: “Visual event-related potentials under external emotional stimuli in bipolar I disorder with and without hypersexuality” by Wang C et al., submitted to Brain Sciences

Dear Authors,

I am very pleased to see that the authors have welcomed my suggestions and have clarified several of the questions I raised in my first round of this review. I believe that this functional study does an excellent work investigating different patterns of visual deficits under different external emotions are impaired in bipolar disorder with (BW) and without (BO) hypersexuality.

I only have two last minor suggestions to do, to further improve the theoretical background of the present article and its argumentation by highlighting how mechanisms underlying impairments in emotion processing are related to frontal lobe dysfunctions. In this regard, I suggest adding evidence from a very recent yet relevant perspective manuscript (https://doi.org/10.17219/acem/146756), in which author focused on providing a deeper understanding of human learning neural networks, particularly on human PFC crucial role, that might also contribute to the advancement of alternative, more precise and individualized treatments for psychiatric disorders, such as bipolar disorder.

Overall, the manuscript contains 5 figures, 3 table and 73 references. This is a timely and needed study, and I look forward to seeing further study on this issue by these authors in the future

Best regards,

Author Response

Please see the attachment, thank you!
